# Effective breastfeeding techniques and associated factors among lactating women in Ethiopia: A systematic review and meta-analysis

Gemeda Wakgari Kitil[1]*, Fikadu Wake Butta[2], Shimelis Tadesse[1], Bekem Dibaba Degefa[1], Gizu Tola Feyisa[1], Addisalem Workie Demsash[2], Adamu Ambachew Shibabaw[2], Shambel Negesse Marami[1], Agmasie Damtew Walle[2], Geleta Nenko Dube[2], Lema Fikadu Wedajo[3], Dejene Edosa Dirirsa[4], Wakuma Wakene Jifar[5], Alex Ayenew Chereka[2]

1 Department of Midwifery, College of Health Sciences, Mattu University, Mattu, Ethiopia, 2 Department of Health Informatics, College of Health Sciences, Mattu University, Mattu, Ethiopia, 3 Department of Midwifery, College of Medicine and Health Sciences, Wallagga University, Nekemte, Ethiopia, 4 Department of Midwifery, College of Health and Medical Sciences, Salale University, Fiche, Ethiopia, 5 Department of Pharmacy, College of Health Sciences, Mattu University, Mattu, Ethiopia

* gemedawa425@gmail.com

## Abstract

### Background

Effective breastfeeding is crucial for maternal and child health, particularly in low-resource settings like Ethiopia. It encompasses a range of skills and strategies, including proper latch, positioning, and frequency of feeding. These techniques not only ensure sufficient milk transfer but also foster bonding between mother and child, enhancing the breastfeeding experience. To effectively prioritize maternal and child health, it is crucial to comprehensively understand the prevalence and factors influencing effective breastfeeding nationwide. Therefore, this study aimed to provide a pooled prevalence of effective breastfeeding techniques and associated factors among lactating mothers in Ethiopia.

### Methods

The study followed the Preferred Reporting Items for Systematic Reviews and Meta-Analysis (PRISMA) checklist, focusing on studies conducted in Ethiopia. We identified eight relevant studies through Google Scholar, Medline, PubMed, Scopus, and the Cochrane Library. Analysis was conducted using STATA version 11, and systematic data extraction employed a checklist to extract relevant data. I2 tests and the Cochrane Q test statistic were used to evaluate heterogeneity. To explore potential publication bias, Egger's weighted regression, Begg's test, and a funnel plot were utilized.

### Results

We identified a total of 955 research articles. Eight studies meeting the eligibility criteria were incorporated into this meta-analysis and systematic review. The pooled prevalence of

**Data Availability Statement:** All the data used and analyzed for this work are publicly available within the paper.

**Funding:** The author(s) received no specific funding for this work.

**Competing interests:** The authors declared that there is no conflict of interest.

effective breastfeeding techniques was 41.99% [95% CI 32.16–51.81]. According to the results of the current meta-analysis, effective breastfeeding techniques were significantly associated with antenatal care follow-up [OR = 1.75, 95% CI 1.10–2.78], maternal educational status [OR = 2.70, 95% CI 1.55–4.71], breastfeeding technique counseling [OR = 2.02, 95% CI 1.41–2.90], the absence of breast problems [OR = 2.26, 95% CI 1.49–3.43], breastfeeding experience [OR = 1.98, 95% CI 1.14–3.46], and immediate skin-to-skin contact [OR = 2.32, 95% CI 1.56–3.44].

## Conclusion

Our findings highlight the vital role of various factors in shaping effective breastfeeding.

## Implications

To improve practices and health outcomes, we recommend targeted interventions, such as strengthening antenatal care, implementing maternal education, and providing comprehensive breastfeeding counseling. Proactively addressing breast problems and prioritizing immediate skin-to-skin contact is crucial for successful breastfeeding.

## 1. Introduction

Breastfeeding technique encompasses the interplay of suckling, position, and attachment as integral components for achieving effective breastfeeding [1]. The act of suckling is a key element in breastfeeding practices and indicates how well the baby expresses milk through latching, rooting, and audible swallowing. This can be explained by the amount and rate of sucking. Position reveals the mother's ability to carry her child on her own body. Similarly, attachment also indicates whether the mother ensures that the majority of the areola is inside the baby's mouth by properly attaching the infant to her breast and nipple [1–4]. The World Health Organization (WHO) recommends employing proper breastfeeding techniques to encourage exclusive breastfeeding, enabling the baby to receive an adequate amount of energy and nutrients. Additionally, breastfeeding protects against a variety of acute and chronic diseases, which is advantageous for the health of both mothers and their infants [5, 6].

Globally, common childhood illnesses claim the lives of approximately 820,000 children, and 20,000 mothers face mortality due to ovarian cancer, breast cancer, type 2 diabetes, and postpartum hemorrhage resulting from ineffective breastfeeding techniques [6–9]. Ineffective breastfeeding contributes to approximately 800,000 infant fatalities annually worldwide [10]. In Ethiopia alone, inadequate breastfeeding is estimated to cause 70,000 newborn deaths each year, accounting for 24 percent of all child mortality. The adoption of effective breastfeeding practices has the potential to prevent these deaths [11].

While effective breastfeeding techniques have shown advantages for both the mother and her child in the short and long term, their implementation in developing countries is currently unsatisfactory [12]. Several studies have proposed a range of solutions, including educating and counseling mothers about proper breastfeeding techniques, providing healthcare professionals with training on appropriate positioning, attachment, and effective suckling, and giving special attention to primipara, young, and low-income women [13–16]. Furthermore, the Integrated Management of Neonatal and Childhood Illness (IMNCI) guidelines in Ethiopia stipulate that healthcare professionals in maternal and child health should ensure that mothers

receive education and counseling on proper breastfeeding techniques. They should also discourage the use of supplemental feedings before six months of age [17].

The Baby-Friendly Hospital Initiative program was jointly launched by the World Health Organization (WHO) and the United Nations Children's Fund (UNICEF) to assist health facilities that provide pregnancy and newborn care services in promoting breastfeeding. It aims to safeguard, encourage, and assist with the implementation of 10 steps to effective breastfeeding. These steps significantly enhance the health of both mothers and newborns, thereby increasing the rate of breastfeeding [9]. More recently, Ethiopia's national guidelines for the treatment of acute malnutrition in children under six months of age, implemented in 2019, included ineffective breastfeeding techniques as a diagnostic criterion for severe acute malnutrition [18]. Additionally, Ethiopia has adopted the WHO IMNCI guidelines, incorporating components for assessing breastfeeding techniques in sick young infants [17].

The prevalence of effective breastfeeding techniques varies across different settings. For instance, in India, it ranged from 18.9% to 43% [19, 20], while in Libya, it is 48% [13]. Furthermore, studies in Nigeria indicate that the practice of effective breastfeeding techniques ranges from 49% to 71.3% [21, 22]. Similarly, in Ethiopia, the practice of effective breastfeeding techniques ranges from 33.2% to 48.0% [23, 24]. Factors such as parity, the age of the mother and child, education level, place of delivery, follow-up on antenatal care (ANC), experience with breastfeeding, maternal occupation, postnatal care (PNC), knowledge of breastfeeding techniques, counseling on breastfeeding throughout prenatal and postpartum care, as well as infant factors like gestational age, the current age of the infant, and bottle feeding, have been identified as influencing the practice of effective breastfeeding techniques [4, 23–26].

Despite several studies reporting on the prevalence and factors associated with effective breastfeeding techniques in Ethiopia, there is currently no nationally representative pooled prevalence of effective breastfeeding techniques and associated factors among lactating mothers. Previous studies have shown diverse, inconsistent, and non-representative results, often limited to specific regions. Therefore, this systematic review and meta-analysis aim to provide nationally representative data on effective breastfeeding techniques and associated factors among lactating mothers in Ethiopia. The conclusions drawn from this research will be beneficial for postnatal care professionals, policymakers, and other concerned entities in recognizing factors contributing to ineffective breastfeeding techniques and addressing or changing them.

## 2. Materials and methods

### 2.1 Source of information and search strategy

To identify published or ongoing studies on the subject, we searched the PROSPERO database and the Database of Abstracts of Reviews of Effects (DARE) at http://www.library.UCSF.edu. The research was conducted using the Preferred Reporting Items for Systematic Reviews and Meta-Analysis (PRISMA) [27] criteria (S1 Table), with the goal of discerning factors associated with effective breastfeeding techniques among Ethiopian women.

The study team developed a review strategy, conducting an online database search from August 1 to October 30, 2023. During the development of search strategies, research articles for this systematic review and meta-analysis were identified using search engines such as Google Scholar, Medline/PubMed, Cochrane Library, Web of Science, Hinari, Science Direct, and ProQuest. Additionally, online national university digital libraries/university repositories like Addis Ababa, Jimma, Debre Birhan, Haramaya University, and the University of Gondar were utilized in the identification process. The reference lists of all included studies were manually searched. The search criteria comprised keywords, free-text search queries, and Medical Subject Headings (MeSH). We incorporated alternative words and employed Boolean operators to

combine them. The search terms we used were: ("magnitude," "prevalence," "determinants," "associated factors," "breastfeeding," and "primipara") AND ("breastfeeding techniques" OR "effective breastfeeding techniques" OR "techniques") AND ("Ethiopia").

The identified papers were directly sent to the Mendeley Citation Manager for inclusion. This systematic review followed the following processes: First, duplicates were eliminated from the electronic database search results by importing them into the Mendeley reference management program. The second stage involved evaluating each article for eligibility using predetermined inclusion and exclusion criteria based on its title, abstract, and full text. Thirdly, a thorough document and manuscript review was carried out, and studies that did not meet the predetermined exclusion criteria were excluded. Lastly, the Joanna Briggs Institute's (JBI) quality assessment technique was used to evaluate the collected publications [28].

## 2.2 Eligibility criteria

This research encompasses studies conducted in Ethiopia, both published and unpublished, that investigate factors associated with effective breastfeeding techniques. The included research in this study comprises full-text articles published in English-language, peer-reviewed journals, as well as master's theses and dissertations that are freely accessible. Exclusion criteria were applied to studies that did not address participants' effective breastfeeding techniques in Ethiopia, were not published in English, had data that was difficult to extract, or included incomplete texts.

The study population (POP) consisted of breastfeeding mothers. The condition (CO) under investigation was effective breastfeeding techniques, and the context (CT) was studies conducted in Ethiopia.

## 2.3 Data extraction and analysis

The Joanna Briggs Institute (JBI) quality rating checklist for cross-sectional studies was used to evaluate the included study's quality (31). Three data extractors (GWK, AAC, and GTF) utilized a standard data extraction checklist to retrieve information from Microsoft Excel.

Initially, reference management software (Mendeley version 1.19.8) was employed to eliminate duplicate articles and consolidate search results from databases. Subsequently, research articles' titles and abstracts were assessed and excluded. Full-text publications served as the basis for evaluating the remaining research articles. The specified inclusion and exclusion criteria were adhered to when assessing the eligibility of primary studies.

The data extraction checklist for the first outcome, which focused on the prevalence of breastfeeding techniques, included the authors' names, year of publication, study location (region), study design, sample size, response rate, and the number of participants who breastfed.

For the second outcome, which focused on factors associated with breastfeeding techniques, data were collected in two-by-two table formats. The log odds ratio (OR) was then calculated using the findings from the original publications. Discrepancies among the four independent reviewers were resolved through additional input from more reviewers (SNM, AAS, FWB, and WWJ) after a discussion to reach a consensus. In cases where the included primary articles lacked sufficient information, corresponding authors of the research articles were contacted by email.

## 2.4 Data analysis and synthesis

We exported the data to STATA Version 11 and used the 95% confidence intervals to compute the pooled effect size. To assess the diversity within the studies considered, we utilized the

Cochran Q test (chi-squared statistic) and the I2 statistic displayed on forest plots. A p-value less than 0.05 was considered statistically significant for Cochran's Q statistical heterogeneity test. As such, the I2 statistic percentiles represent varying levels of heterogeneity: zero for none, twenty-five for low, fifty for moderate, and seventy-five for high. The I2 statistic ranges from 0% to 100%. We assessed publication bias using a funnel plot, where asymmetry could indicate bias. Additionally, we employed Begg's test and Egger's weighted regression to further assess publication bias. A significance level of $P < 0.05$ was adopted to determine the statistical significance of publication bias.

## 2.5 Quality assessment and appraisal

A standardized tool was employed to categorize bias potential and elucidate variations in the findings of the included research, aiming to assess the quality of each study. Three independent reviewers conducted a quality control check. Additionally, the authors adapted the Newcastle-Ottawa Scale (NOS), a suitable instrument for assessing bias risk in observational studies, to evaluate the methodological and other aspects of each publication. After analyzing the publications, we determined that those with a modified NOS component score of seven or higher were deemed relevant.

## 2.6 Patient and public involvement

No patients were involved in developing the research question, determining outcome measures, designing the study, recruiting participants, analyzing data, interpreting results, or implementing the research. Moreover, the research design did not involve direct public participation, and there is no plan to disseminate the results to patients.

## 2.7 Ethical consideration

As part of this systematic review, we applied the Preferred Reporting Items for Systematic Reviews and Meta-Analysis (PRISMA) criteria to evaluate the literature. To address potential conflicts of interest, as well as voice and representation issues, we systematically read and acknowledged the included research in the manuscript. Therefore, the dates of participant recruitment and/or medical record access are not relevant, and ethics approval is not applicable.

## 3. Result

### 3.1. Study selections

A total of 955 articles were identified through various databases. After reviewing titles and abstracts, 600 duplicate articles were identified and subsequently removed from the analysis. Additionally, 330 research publications deemed irrelevant were also excluded. Following a review of the entire of the remaining articles' text, 17 studies were excluded because they did not comply with the predetermined eligibility criteria. The final eight research articles were included in the final meta-analysis and systematic review (See Fig 1).

### 3.2 Characteristics of the included studies

The final meta-analysis and systematic review comprised eight studies carried out in Ethiopia. The study with the largest sample size (786 participants) was in the Amhara region, Gidan district [26], while the study with the lowest sample size (253 participants) was conducted in Addis Ababa [29]. All studies incorporated into the analysis were cross-sectional [4, 23–26, 29–31].

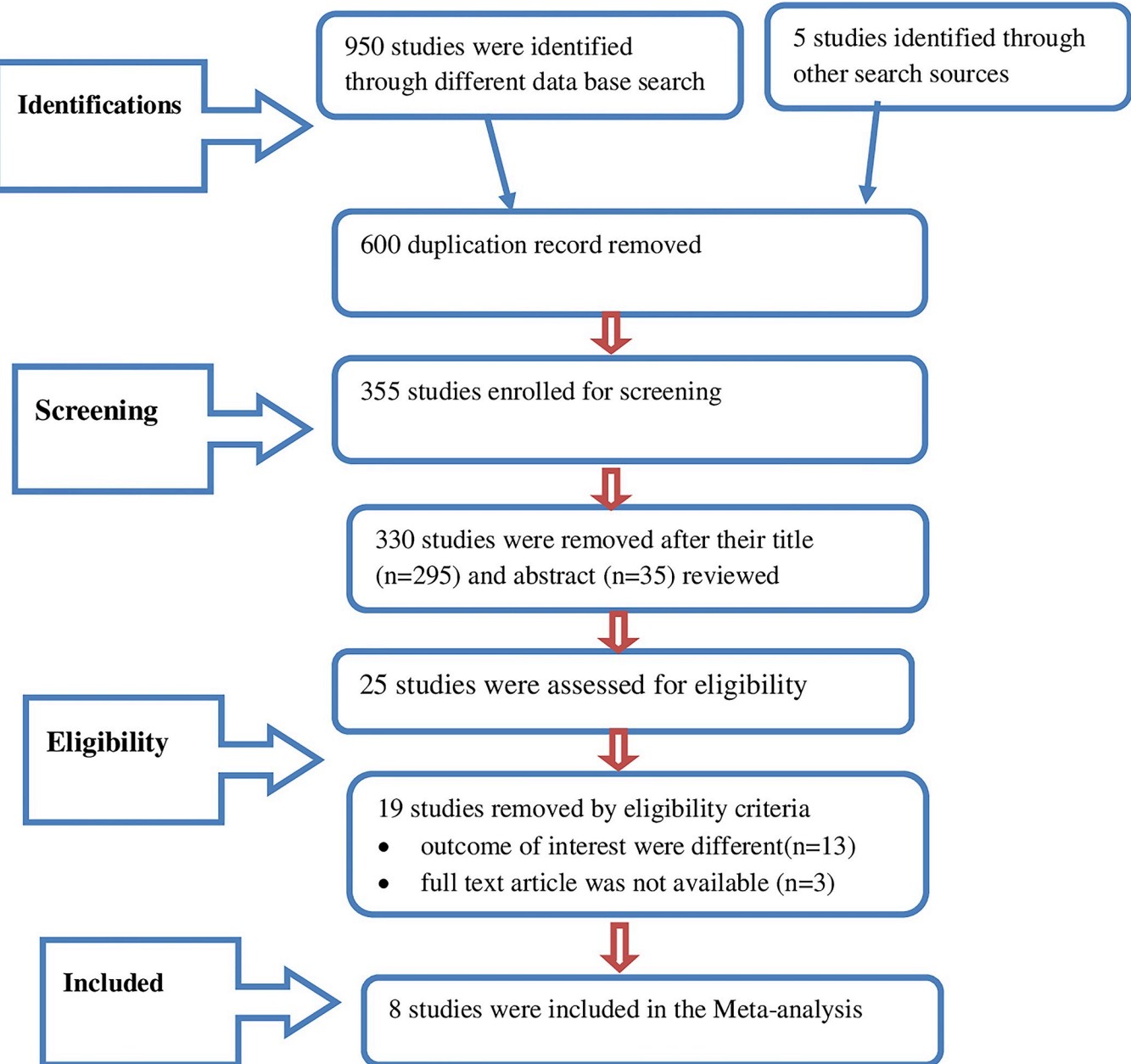

**Fig 1. PRISMA flow diagram showing systematic review and meta-analysis selection process of articles.**

In terms of study settings, four studies were performed in the Amhara region [23, 26, 30], two in SNNPR [25, 31], one in Addis Ababa [29], and one in Harar city [4] (See Table 1).

### 3.3 Prevalence of effective breastfeeding techniques

The pooled prevalence of effective breastfeeding techniques was 41.99% (95% CI: 32.16–51.81). No heterogeneity was observed among the incorporated studies (I2 = 0.0, p = 0.995). The highest prevalence of effective breastfeeding techniques (48.0%, 95% CI: 19.73–76.27) was found in Gondar town [23], while the lowest prevalence (33.2%, 95% CI: 6.56–59.84) was reported in East Gojjam Sinan werada [24] (See Fig 2).

**Table 1. Descriptive summary of eight studies included in the 2023 meta-analysis on factors affecting effective breastfeeding techniques among women in Ethiopia.**

| Authors | Year | Region | Study area | Study Design | Quality score | study population | Response rate | Sample Size | Prevalence |
|---|---|---|---|---|---|---|---|---|---|
| Asmamaw et.al [26] | 2022 | Amhara | Gidan district | cross-sectional | 9 | Lactating mothers | 96.7% | 786 | 42.9 |
| Safayi et.al [23] | 2020 | Amhara | Gondar town | cross-sectional | 8 | lactating mothers | 99% | 414 | 48.0 |
| Tiruye et.al [4] | 2018 | Harar | Harar city | cross-sectional | 9 | lactating mothers | 97.6% | 422 | 43.4 |
| Yilak et al [25] | 2020 | SNNPR | South Ari district | cross-sectional | 8 | lactating mothers | 99.8% | 415 | 36.5 |
| Mekurya et al [29] | 2020 | Addis Ababa | Health centers | cross-sectional | 7 | lactating mothers | 100% | 253 | 40.3 |
| Alemie et al [24] | 2023 | Amhara | Sinan Woreda | cross-sectional | 9 | lactating mothers | 99.2% | 389 | 33.2 |
| Yehualashet [30] | 2022 | Amhara | Basona district | cross-sectional | 8 | lactating primipara mothers | 86.5% | 415 | 46.2 |
| Jawaro [31] | 2023 | SNNPR | Arba Minch town | cross-sectional | 7 | lactating primipara mothers | 100% | 407 | 46.9 |

## 3.4 Publication bias

We utilized both a visual examination of the asymmetry in a funnel plot and Egger's regression test to determine if publication bias was present. Although the funnel plot displayed an uneven distribution upon initial observation, the outcome of Egger's test (P = 0.32) did not achieve statistical significance. The symmetrical distribution of the funnel plot indicated the lack of publication bias, while an asymmetric distribution might hint at bias upon visual examination. Additionally, the lack of publication bias yielded a statistically significant result, as indicated by Egger's test (See Fig 3).

## 3.5 Factors associated with effective breastfeeding techniques

Eight papers were included in the meta-analysis to determine factors influencing effective breastfeeding techniques collectively. Utilizing the command 'metan logor selogor, xlab(0.1, 1, 10) label(namevar = authors) by (factors) random texts(180) eform,' we evaluated the pooled effects of the odds ratio.

Mothers who attended ANC follow-ups demonstrated a 1.75-fold higher likelihood of employing effective breastfeeding techniques compared to their counterparts [OR = 1.75, 95% CI: 1.10, 2.78]. Similarly, mothers with at least a primary educational level were 2.70 times more likely to practice effective breastfeeding techniques compared to those with no formal education [OR = 2.70, 95% CI: 1.55, 4.71].

When it comes to counseling on breastfeeding techniques, mothers who received guidance on effectiveness were 2.02 times more likely to adopt effective breastfeeding practices compared to those who did not receive such counseling [OR = 2.02, 95% CI: 1.41, 2.90]. In the context of this study, mothers with prior breastfeeding experience were 1.98 times more likely to adopt effective breastfeeding techniques compared to those without previous experience [OR = 1.98, 95% CI: 1.14–3.46].

Furthermore, mothers without breast problems had a 2.26 times higher likelihood of practicing effective breastfeeding techniques compared to those who experienced such issues [OR = 2.26, 95% CI: 1.49, 3.43]. Lastly, mothers who experienced immediate skin-to-skin contact with their newborns after birth were 2.32 times more likely to practice effective breastfeeding techniques compared to those without immediate contact [OR = 2.32, 95% CI: 1.56, 3.44] (See Fig 4).

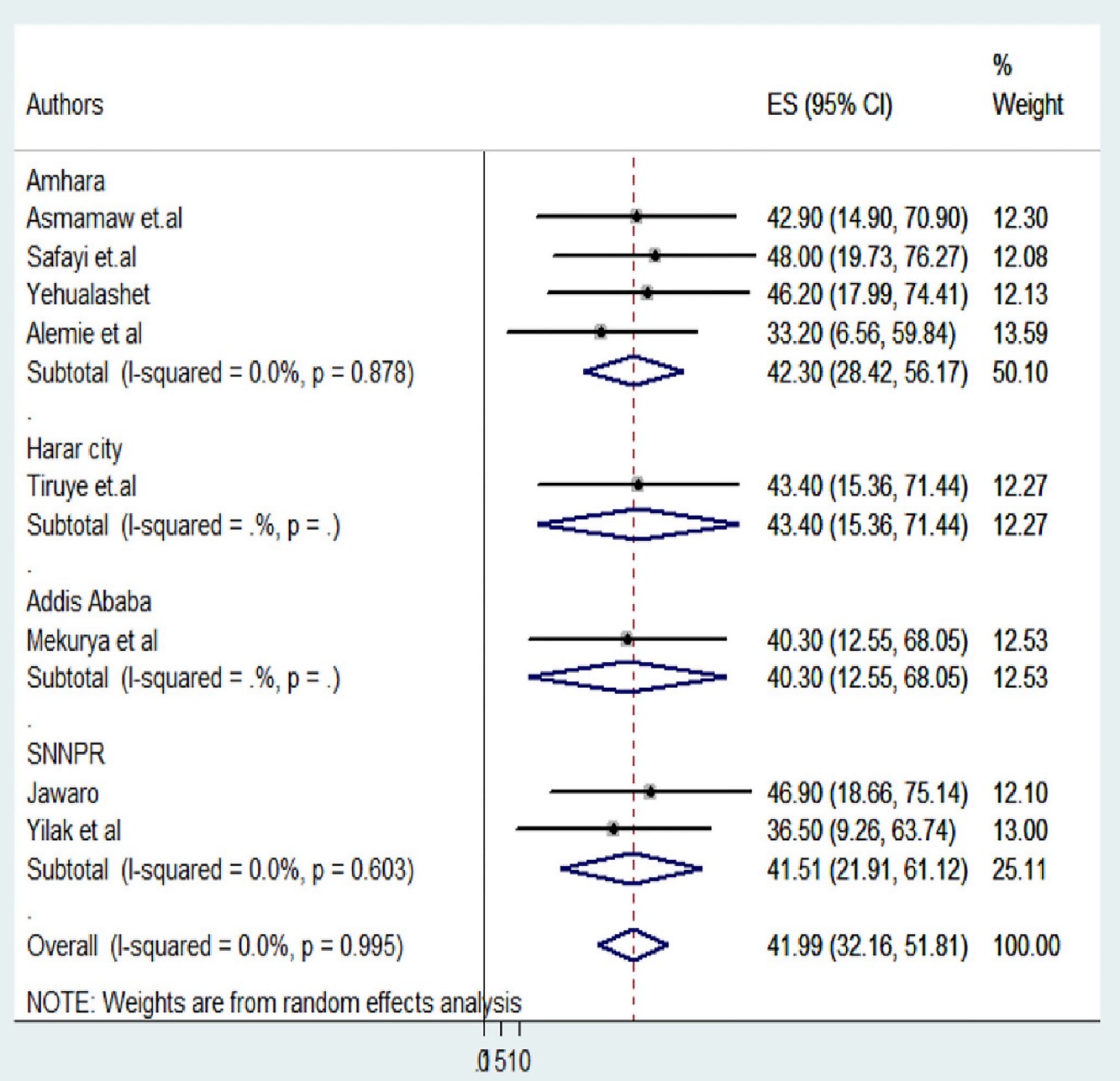

**Fig 2. Forest plot of the pooled prevalence of breastfeeding techniques among lactating mothers in Ethiopia, 2023.**

## 4. Discussion

Based on the available information, this meta-analysis and systematic review represent the first of their kind conducted at the national level to evaluate the prevalence and factors associated with effective breastfeeding techniques among lactating mothers in Ethiopia. The pooled prevalence of effective breastfeeding techniques in Ethiopia is 41.99% (95% CI: 32.16–51.81), indicating a substantial variation in the adoption of these techniques across studies. Despite the absence of a meta-analysis on this research subject, the effective breastfeeding technique

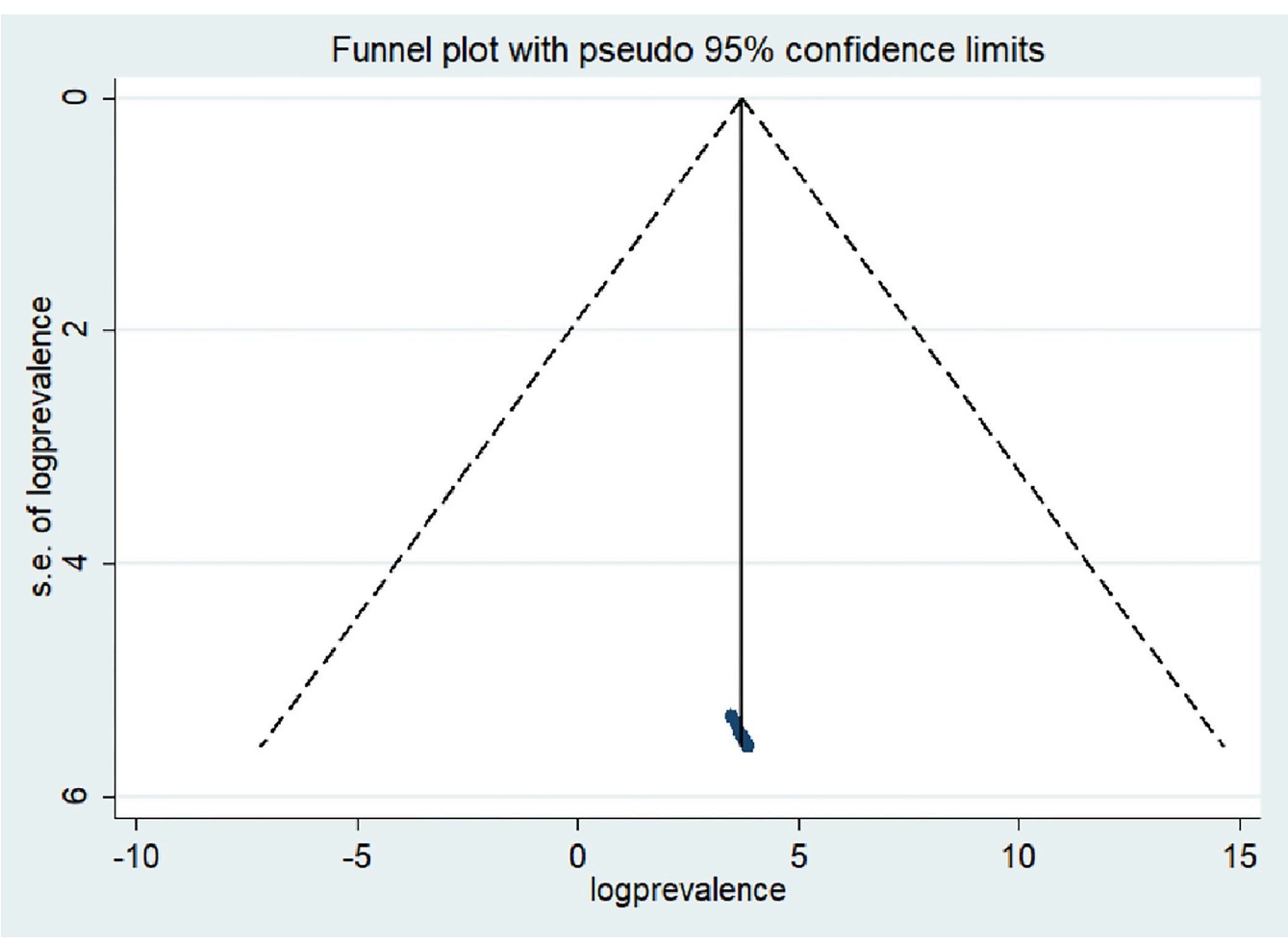

**Fig 3. Funnel plot of included studies to test publications bias in Ethiopia, 2023.**

reported in this study is consistent with findings from other countries such as India (43%) [20], Libya (48%) [13], and Nigeria (41%) [21].

The findings of this systematic review and meta-analysis indicate a lower prevalence compared to the Nigerian study (71.3%) [22] and the Denmark study (52%) [14]. Possible explanations for this discrepancy may stem from variations between the local and national study populations. Other contributing factors could include a combination of cultural, economic, and healthcare influences.

One of the key findings in this meta-analysis is the significant association between effective breastfeeding techniques and antenatal care follow-up. Mothers who underwent antenatal care were 1.75 times more likely to practice effective breastfeeding techniques compared to those without such follow-up (OR = 1.75, 95% CI: 1.10, 2.78). This result aligns with research carried out in the southeastern region of Nigeria [21]. The potential justification is that mothers attending antenatal care sessions are likely exposed to information and guidance on breastfeeding, fostering a better understanding of techniques and benefits. Comprehensive health monitoring during antenatal care visits may aid in the early identification and resolution of potential breastfeeding challenges.

Maternal educational status emerged as a crucial factor influencing effective breastfeeding techniques. Mothers with at least a primary education exhibited a 2.70 times higher likelihood

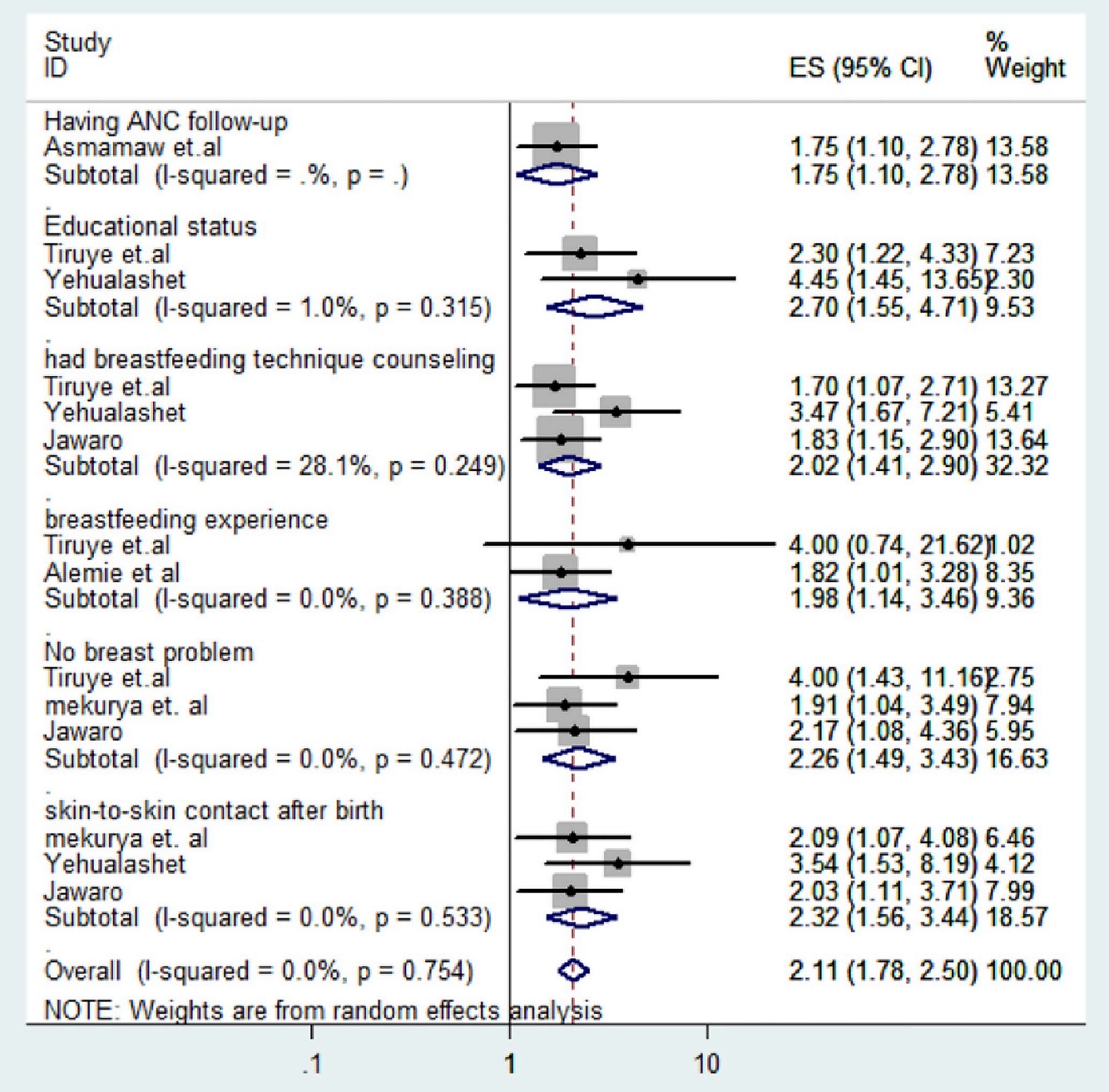

**Fig 4. Forest plot of associated factors of effective breastfeeding techniques among lactating mothers in Ethiopia, 2023.**

of employing these techniques compared to those with no formal education (OR = 2.70, 95% CI 1.55, 4.71). This finding aligns with research conducted in Sri Lanka [20], Saudi Heraa General Hospital [32], and West Bengal Kolkata Hospital [15]. One plausible justification is that education equips mothers with knowledge and awareness, fostering a deeper understanding of the nutritional and health benefits associated with proper breastfeeding practices.

The study's findings reveal that mothers with prior breastfeeding experience were 1.98 times more likely to adopt effective breastfeeding techniques compared to those without

previous experience (OR = 1.98, 95% CI: 1.14–3.46). This finding aligns with a study conducted in India, specifically in East Delhi [33], and in western Denmark [14]. Possible explanations suggest that accumulated knowledge, skills, and confidence from previous experiences can significantly contribute to the successful practice of breastfeeding.

Furthermore, our analysis revealed a positive correlation between effective breastfeeding techniques and counseling on breastfeeding. Mothers who received counseling were 2.02 times more likely to practice effective breastfeeding than their counterparts (OR = 2.02, 95% CI 1.41, 2.90). This finding aligns with several studies conducted in rural areas of Nagpur district [34], North India [35], Bangladesh [16], and Saudi Heraa General Hospital [32]. The observed correlation can be attributed to the provision of valuable information and education, which, when combined with the development of problem-solving skills and increased maternal confidence, likely empowers mothers to adopt and sustain proper breastfeeding techniques.

Another significant factor contributing to effective breastfeeding practices is the absence of breast problems. Mothers without breast problems had a 2.26 times higher likelihood of employing effective techniques (OR = 2.26, 95% CI 1.49, 3.43), as supported by a study conducted in Denmark [14]. This could be because mothers experiencing no breast issues are more likely to maintain optimal breast health, thereby facilitating easier and more comfortable breastfeeding experiences. Pain, discomfort, or complications related to breast problems could hinder a mother's ability to practice effective breastfeeding techniques.

The present study demonstrated that mothers who practiced immediate skin-to-skin contact with their infants were 2.32 times more likely to adopt effective breastfeeding practices compared to those who did not engage in such contact (OR = 2.32, 95% CI 1.56, 3.44). This result is in line with research done in Colombia [36]. One possible explanation for this correlation is that immediate skin-to-skin contact promotes effective breastfeeding by fostering bonding, releasing hormones like oxytocin, regulating the baby's temperature, and encouraging proper latch. This early connection increases the likelihood that mothers will overcome challenges and develop successful breastfeeding practices.

In conclusion, our meta-analysis provides valuable insights into the prevalence of effective breastfeeding techniques and the factors associated with their adoption. These findings can inform public health initiatives and healthcare strategies aimed at improving breastfeeding outcomes and maternal-infant health.

### 4.1. Limitations of the study

The study has limitations, notably the small sample sizes of the included studies. Additionally, all the reviewed studies adopted cross-sectional designs, preventing the identification of causal relationships.

## 5. Conclusion

In conclusion, our comprehensive meta-analysis reveals a low prevalence of effective breastfeeding techniques. Crucial influencers identified include antenatal care follow-up, maternal educational status, breastfeeding technique counseling, absence of breast problems, and immediate skin-to-skin contact, each playing a unique role in shaping successful breastfeeding behaviors.

## 6. Recommendation

To enhance breastfeeding practices in Ethiopia, it is recommended to strengthen antenatal care programs for pregnant mothers, with a particular emphasis on breastfeeding education.

Implement targeted maternal education initiatives to empower mothers with knowledge and decision-making skills. Expand and promote breastfeeding counseling services to provide tailored guidance and support. Proactively address and manage breast problems, ensuring timely intervention and assistance. Prioritize and facilitate immediate skin-to-skin contact between mothers and newborns to enhance breastfeeding initiation.

## Supporting information

**S1 Table. A diagram illustrating the PRISMA 2020 checklists of systematic review and meta-analysis of effective breastfeeding techniques among women in Ethiopia.**
(DOCX)

**S1 Data. Data set for prevalence.**
(CSV)

**S2 Data. Data set for factors.**
(CSV)

## Acknowledgments

The authors extend their gratitude to the authors of the included primary studies, which were used as a source of information to conduct this systematic review and meta-analysis.

## Author Contributions

**Conceptualization:** Gemeda Wakgari Kitil, Fikadu Wake Butta, Bekem Dibaba Degefa, Gizu Tola Feyisa, Addisalem Workie Demsash, Adamu Ambachew Shibabaw, Shambel Negesse Marami, Agmasie Damtew Walle, Lema Fikadu Wedajo, Dejene Edosa Dirirsa, Wakuma Wakene Jifar, Alex Ayenew Chereka.

**Data curation:** Gemeda Wakgari Kitil, Shimelis Tadesse, Gizu Tola Feyisa, Geleta Nenko Dube, Wakuma Wakene Jifar.

**Formal analysis:** Gemeda Wakgari Kitil, Fikadu Wake Butta, Bekem Dibaba Degefa, Addisalem Workie Demsash, Shambel Negesse Marami, Lema Fikadu Wedajo, Alex Ayenew Chereka.

**Funding acquisition:** Gemeda Wakgari Kitil.

**Investigation:** Gemeda Wakgari Kitil, Fikadu Wake Butta, Shimelis Tadesse, Gizu Tola Feyisa, Addisalem Workie Demsash, Adamu Ambachew Shibabaw, Agmasie Damtew Walle, Wakuma Wakene Jifar.

**Methodology:** Gemeda Wakgari Kitil, Fikadu Wake Butta, Shimelis Tadesse, Bekem Dibaba Degefa, Adamu Ambachew Shibabaw, Shambel Negesse Marami, Geleta Nenko Dube, Dejene Edosa Dirirsa, Alex Ayenew Chereka.

**Project administration:** Gemeda Wakgari Kitil, Bekem Dibaba Degefa, Gizu Tola Feyisa, Addisalem Workie Demsash.

**Resources:** Gemeda Wakgari Kitil, Lema Fikadu Wedajo.

**Software:** Gemeda Wakgari Kitil, Shambel Negesse Marami, Geleta Nenko Dube, Wakuma Wakene Jifar, Alex Ayenew Chereka.

**Supervision:** Gemeda Wakgari Kitil, Fikadu Wake Butta, Shimelis Tadesse, Bekem Dibaba Degefa, Gizu Tola Feyisa, Addisalem Workie Demsash, Shambel Negesse Marami, Agmasie

Damtew Walle, Lema Fikadu Wedajo, Dejene Edosa Dirirsa, Wakuma Wakene Jifar, Alex Ayenew Chereka.

**Validation:** Gemeda Wakgari Kitil, Fikadu Wake Butta, Shimelis Tadesse, Adamu Ambachew Shibabaw, Geleta Nenko Dube, Dejene Edosa Dirirsa, Wakuma Wakene Jifar, Alex Ayenew Chereka.

**Visualization:** Gemeda Wakgari Kitil, Bekem Dibaba Degefa, Agmasie Damtew Walle, Geleta Nenko Dube, Dejene Edosa Dirirsa, Wakuma Wakene Jifar.

**Writing – original draft:** Gemeda Wakgari Kitil, Shimelis Tadesse, Bekem Dibaba Degefa, Gizu Tola Feyisa, Adamu Ambachew Shibabaw, Shambel Negesse Marami, Dejene Edosa Dirirsa, Wakuma Wakene Jifar, Alex Ayenew Chereka.

**Writing – review & editing:** Gemeda Wakgari Kitil, Fikadu Wake Butta, Addisalem Workie Demsash, Shambel Negesse Marami, Agmasie Damtew Walle, Lema Fikadu Wedajo, Alex Ayenew Chereka.

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
