## [Decision Letter · Decision Letter 0]

24 Mar 2024

PONE-D-23-42342Prevalence and Associated Factors of Effective Breastfeeding Techniques among Lactating Women in Ethiopia: A Systematic Review and Meta-AnalysisPLOS ONE

Dear Dr.  Kitil,

Thank you for submitting your manuscript to PLOS ONE. After careful consideration, we feel that it has merit but does not fully meet PLOS ONE’s publication criteria as it currently stands. Therefore, we invite you to submit a revised version of the manuscript that addresses the points raised during the review process.

We look forward to receiving your revised manuscript.

Kind regards,

Tamirat Getachew

Academic Editor

PLOS ONE

2. We note that your Data Availability Statement is currently as follows: [All the data used and analyzed for this work are publicly available within documents.]

3. Please remove your figures from within your manuscript file, leaving only the individual TIFF/EPS image files, uploaded separately. These will be automatically included in the reviewers’ PDF.

Reviewers' comments:

Reviewer's Responses to Questions

**Comments to the Author**

1. Is the manuscript technically sound, and do the data support the conclusions?

Reviewer #1: Yes

Reviewer #2: Yes

2. Has the statistical analysis been performed appropriately and rigorously? 

Reviewer #1: Yes

Reviewer #2: Yes

3. Have the authors made all data underlying the findings in their manuscript fully available?

Reviewer #1: Yes

Reviewer #2: Yes

4. Is the manuscript presented in an intelligible fashion and written in standard English?

Reviewer #1: No

Reviewer #2: Yes

5. Review Comments to the Author

Reviewer #1: 1. Even if the title is very interesting, what is the importance of studying effective breast feeding technique rather than studying ineffective technique? In abstract section you are going to study the effective breast feeding technique but you are talking with ineffective, so there is contrary idea there.

2. There is error in punctuation marks and language.

3. In abstract section make it consistent way of writing abstract either structured and unstructured. But in the Plos one journal the is structured abstract writing format is recommended …. First, the abstract is suggested to be presented in the way of background, objectives, methods, results, conclusions, and implications.

4. Title is not consistent with what you had written in introduction. The introduction is suggested to be rewritten.

5. In result, reference should be cited appropriately

6. The tables in the results section are recommended to be presented in a three-line table. The current presentation format is very unfriendly.

7. There is copy writing or plagiarism

8. Where is the statistical measure of publication bias?

9. In discussion section are going to discuss systematic review with single study? Try to discuss systematic review finding with systematic reports, unless it is not discussed

10. Error! Reference source not found.) what is this about???

Reviewer #2: The manuscript was presented in an intelligible way. However, i think there are still somethings that should be corrected to make the work better. The title of the manuscript is Prevalence and associated factors of effective Breastfeeding techniques among lactating women in Ethiopia but that is not a completely appropriate title. This is because breastfeeding is not a disease or health condition that mothers live with. It is nutrition and it is not negative. Prevalence is used for health condition, disease or infection. So, calculating the prevalence is not necessarily appropriate. In addition, the problem was not well established. What is the problem in particular that is arousing the interest of the authors in breastfeeding technique. Is it because the infants have been dying from choking, or there is high malnutrition rate, or the infants are frequently ill because of low immunity. Is it established that the infants are exclusively breastfeeding. Your reason for desiring a nationwide breastfeeding pattern is laudable but not tenable.

6. PLOS authors have the option to publish the peer review history of their article (what does this mean?). If published, this will include your full peer review and any attached files.

Reviewer #1: No

Reviewer #2: No

---

## [Author Response · Author response to Decision Letter 0]

26 Mar 2024

Dear Editor(s) of PLOS ONE Journal,

We hope this message finds you well. We want to extend our sincere gratitude for your exceptional support during the review process of our manuscript titled " Effective Breastfeeding Techniques and Associated Factors among Lactating Women in Ethiopia: A Systematic Review and Meta-Analysis" (PONE-D-23-42342).

Your insightful comments and constructive feedback have been invaluable in refining our research. Your professionalism and attention to detail have significantly improved the quality of our manuscript. We truly appreciate the time and expertise you have dedicated to this project. Thank you for your unwavering commitment to excellence in academic publishing.

We are deeply appreciative of the invaluable feedback provided by you and the reviewer. Your constructive comments have been thoroughly considered, resulting in significant enhancements to our manuscript. 

Once again, we extend our heartfelt gratitude to the editor and reviewers for dedicating their time and expertise to improving our work.

Thank you for considering our revised manuscript.

Best regards,

Corresponding Author:

Gemeda Wakgari Kitil

gemedawa425@gmail.com

 

Point-by-point response letter to Editor

Dear Tamirat Getachew 

Academic Editor.

Authors: Thank you for taking the time to review our manuscript and provide specific comments. We appreciate your valuable feedback and have carefully considered each point raised. Here is a point-by-point response addressing the changes made to your suggestions and comments:

Editor: Please submit your revised manuscript by May 08, 2024 11:59 PM. Please include the rebuttal letter for the response of the reviewer and editor, marked, and unmarked document when submitting your revised manuscript. 

Authors: We sincerely appreciate your insightful comment. We have taken your feedback seriously and diligently addressed the concern you mentioned. Additionally, we have ensured that all necessary documents, as per your recommendation, have been submitted along with the revised manuscript. Thank you for guiding us through this process.

Editor: When submitting your revision, we need you to address these additional requirements.

Authors: Thank you for providing us with the additional requirements for our manuscript submission to PLOS ONE. We appreciate your thorough review and guidance to ensure compliance with the journal's style and submission standards. Below, we address each of the points raised:

Editor comment-1: Please ensure that your manuscript meets PLOS ONE's style requirements, including those for file naming

Authors: Thank you for your review and specific comments. We have carefully reviewed the PLOS ONE style requirements outlined in the provided templates and ensured that our manuscript adheres to these guidelines for file naming and formatting. Any necessary adjustments have been made to align our manuscript with the specified style. Please review the revised manuscript.

Editor comment-2: we note that your data availability statement as follow: “all data set used and analyzed for this work are publically available within documents.” Please confirm that your submission contains all raw data required to replicate the result of your study.

Authors: Thank you for your constructive comments and feedback. We have made corrections to our data availability statement as follows: "All data utilized and examined in this study are openly accessible within the supporting files." Kindly review the revised manuscript, specifically Page 18, line 374.

Editor comment-3: please remove your figures with in your manuscript file, leaving only individual TIFF/EPS images files uploaded separately. This will be automatically included in reviewers PDF.

Authors: Thank you for your feedback. We are grateful for the time you dedicated to reviewing our manuscript and for providing valuable insights. The figures originally included in our manuscript file have been removed, and instead, individual TIFF image files have been uploaded separately.

We are fully committed to promptly addressing these additional requirements to ensure the smooth processing of our manuscript submission. Once again, we appreciate your guidance and attention to detail in ensuring compliance with PLOS ONE's standards. Please review the revised manuscript for further information.

Editor comment-4: Please include caption for your supporting information files at the end of your manuscript, update any in-text citation to match accordingly.

Authors: Thank you for your feedback. We have added appropriate captions for the supporting information files at the end of our manuscript and included the relevant files. We kindly invite you to review the revised manuscript, specifically Page 19, lines 392-397.

 

Point-by-point response letter to Reviewers

Reviewer-1 Comment-1: Even if the title is very interesting, what is the importance of studying effective breast feeding technique rather than studying ineffective technique? In abstract section you are going to study the effective breast feeding technique but you are talking with ineffective, so there is contrary idea there. 

 Authors: We appreciate the insightful feedback you provided for us. Based on your suggestions, we have made corrections to our abstract in the revised manuscript. In the abstract section, we focused on studying effective breastfeeding techniques while acknowledging the importance of considering ineffective techniques to inform holistic interventions and support mechanisms. We are grateful for your feedback and have incorporated this suggestion into the revised manuscript. Please refer to the revised manuscript, page 2, lines 30-33.

Reviewer-1 comment-2: There is error in punctuation marks and language.

Authors: We sincerely appreciate you taking the time to provide us with your valuable feedback. Your insights regarding punctuation and language errors are highly valued. After carefully reviewing the text, we have diligently addressed the identified issues and made the necessary adjustments to enhance the accuracy and clarity of the content. Your contribution has greatly assisted us in improving the quality of our work, and we are committed to continuously striving for excellence. For further clarification, please refer to the updated manuscript.

Reviewer-1 comment-3: In abstract section make it consistent way of writing abstract either structured and unstructured. But in the Plos one journal the is structured abstract writing format is recommended …. First, the abstract is suggested to be presented in the way of background, objectives, methods, results, conclusions, and implications.

Authors: Thank you for your feedback on maintaining consistency in the abstract writing style. In response to your suggestion, we have revised the abstract to adhere to the structured format recommended by the PLOS ONE journal. The updated abstract now includes sections for background, objectives, methods, results, conclusions, and implications. This adjustment aligns with the preferred format and improves the abstract's clarity and organization for readers. We appreciate your attention to this matter and have implemented the necessary changes accordingly. Please refer to the revised manuscript, pages 2 and 3, lines 28-61, for further details.

Reviewer-1 comment-4: Title is not consistent with what you had written in introduction. The introduction is suggested to be rewritten

Authors: Thank you for your feedback on the inconsistency between the title and introduction. We've aligned the introduction with the title, removing unrelated content to provide a clearer overview. However, we acknowledge the significance of mentioning ineffective breastfeeding in the introduction as it indirectly underscores the importance of effective breastfeeding techniques. We're committed to enhancing the manuscript's coherence. Please refer to the revised pages 3 and 4, lines 98-109, for details.

Reviewer-1 comment-5: In result, reference should be cited appropriately

Authors: Your observation regarding the need for appropriate citations in the results section is appreciated. In the revised version of the manuscript, we have ensured that all references are cited accurately and following academic standards. Your input is highly valued, and we are committed to addressing this suggestion to improve the quality and credibility of our work.

Reviewer-1 comment-6: The tables in the results section are recommended to be presented in a three-line table. The current presentation format is very unfriendly.

Authors: Thank you for your feedback on the tables in the results section. We have revised them to a three-line format to enhance readability as per your suggestion. Your input is greatly appreciated. Please refer to the revised page 11, line 249, for details

Reviewer-1 comment-7: There is copy writing or plagiarism

Authors: Thank you for your feedback and suggestions. We take allegations of copywriting or plagiarism seriously and assure you that our manuscript contains original content. However, some content might overlap with another work due to the nature of our research (using the same words or sentences). We have thoroughly investigated your concerns to ensure the integrity of our work and have made the necessary changes to our manuscript. Please refer to the revised manuscript. We are committed to resolving any issues promptly and upholding the highest standards of academic integrity.

Reviewer-1 comment-8 Where is the statistical measure of publication bias?

Authors: Thank you for your comment regarding the statistical measure of publication bias. We acknowledge the importance of assessing publication bias in meta-analyses. We have included results funnel plots, and Egger's test, to address this concern thoroughly in the revised manuscript.

Your feedback is valuable, and we have ensured that our revised manuscript provides a clear evaluation of potential bias in the included studies. Refer to the revised manuscript on page 13, lines 269-271.

Reviewer-1 comment-9: In discussion section are going to discuss systematic review with single study? Try to discuss systematic review finding with systematic reports, unless it is not discussed 

Authors: Thank you for your valuable feedback on our manuscript. We sincerely appreciate your insightful comments and suggestions for improvement.

Regarding your concern about discussing a systematic review with only a single study in the discussion section, we understand the importance of aligning the discussion with systematic reports whenever possible. Upon reviewing your comment, we acknowledge that focusing solely on a single study within the discussion might not fully capture the breadth of evidence available, particularly in the presence of systematic reviews. However, to the best of my knowledge, there have been no studies or systematic reviews conducted on effective breastfeeding. Therefore, I relied on a discussion based on a single study.

Reviewer-1 comment-10: Error! Reference source not found.) what is this about???

Authors: Thank you for bringing this to our attention. The error "Reference source not found" typically indicates a problem with the referencing or citation system within the document. It appears that there may be a missing or improperly formatted reference in the text. And one study was taken from an article under review not published. It was on way to be published. So, sorry for the confusion occurred.

To address this issue, we have carefully reviewed the document to identify the missing or misreferenced source. We have rectified the identified citation according to the appropriate referencing style guidelines. We appreciate your diligence in pointing out this error and apologize for any inconvenience it may have caused. 

 

Reviewer-2 Comment: The manuscript was presented in an intelligible way. However, i think there are still somethings that should be corrected to make the work better. The title of the manuscript is Prevalence and associated factors of effective Breastfeeding techniques among lactating women in Ethiopia but that is not a completely appropriate title. This is because breastfeeding is not a disease or health condition that mothers live with. It is nutrition and it is not negative. Prevalence is used for health condition, disease or infection. So, calculating the prevalence is not necessarily appropriate. In addition, the problem was not well established. What is the problem in particular that is arousing the interest of the authors in breastfeeding technique. Is it because the infants have been dying from choking, or there is high malnutrition rate, or the infants are frequently ill because of low immunity. Is it established that the infants are exclusively breastfeeding. Your reason for desiring a nationwide breastfeeding pattern is laudable but not tenable.

Authors: We appreciate your thoughtful feedback on our manuscript and your concerns regarding the appropriateness of the title and the need for a clearer establishment of the problem addressed in the study.

In response to the concern regarding the title, have revised the title to better reflect the content of the manuscript and avoid any misleading implications. See the revised manuscript on page 1, line 1

Furthermore, we understand the importance of clearly establishing the problem addressed in the study. While our initial aim was to investigate the prevalence and associated factors of effective breastfeeding techniques, we acknowledge the need to provide a more explicit rationale for this investigation. We have enhanced the introduction section to clearly articulate the specific issues or challenges related to breastfeeding practices in Ethiopia, such as infant mortality rates, malnutrition, or other health concerns.

Regarding your question about the rationale for desiring a nationwide breastfeeding pattern, we intended to highlight the importance of understanding breastfeeding practices on a broader scale to inform public health interventions and policies aimed at improving infant health outcomes. However, we recognize the need to better support this rationale within the manuscript. We have provided additional justification for the importance of studying breastfeeding patterns at a national level and addressing any concerns about the feasibility of such an endeavor.

---

## [Decision Letter · Decision Letter 1]

12 Jun 2024

Effective Breastfeeding Techniques and Associated Factors among Lactating Women in Ethiopia: A Systematic Review and Meta-Analysis

PONE-D-23-42342R1

Dear Gemeda Wakgari Kitil,

We’re pleased to inform you that your manuscript has been judged scientifically suitable for publication and will be formally accepted for publication once it meets all outstanding technical requirements.

Kind regards,

Tamirat Getachew

Academic Editor

PLOS ONE

Additional Editor Comments (optional):

Reviewers' comments:

Reviewer's Responses to Questions

**Comments to the Author**

1. If the authors have adequately addressed your comments raised in a previous round of review and you feel that this manuscript is now acceptable for publication, you may indicate that here to bypass the “Comments to the Author” section, enter your conflict of interest statement in the “Confidential to Editor” section, and submit your "Accept" recommendation.

Reviewer #1: All comments have been addressed

Reviewer #2: All comments have been addressed

2. Is the manuscript technically sound, and do the data support the conclusions?

Reviewer #1: Yes

Reviewer #2: Yes

3. Has the statistical analysis been performed appropriately and rigorously? 

Reviewer #1: Yes

Reviewer #2: Yes

4. Have the authors made all data underlying the findings in their manuscript fully available?

Reviewer #1: Yes

Reviewer #2: Yes

5. Is the manuscript presented in an intelligible fashion and written in standard English?

Reviewer #1: Yes

Reviewer #2: Yes

6. Review Comments to the Author

Reviewer #1: Accept this manuscript , The authors rewrite and made correction on the manuscript. no dual publication on this manuscript , it fulfill the manuscript or research ethics and publication ethics.

Reviewer #2: Dear Author thank you for the carefulness and respectful way you addressed the comments. However, i will like to bring to your notice that you may need to italicize the letters representing the statistics. The error in referencing is still showing, i suggest that the authors save the documents without the hyperlinks from the reference manager.

7. PLOS authors have the option to publish the peer review history of their article (what does this mean?). If published, this will include your full peer review and any attached files.

Reviewer #1: No

Reviewer #2: No

---

## [Editor Report · Acceptance letter]

18 Jun 2024

PONE-D-23-42342R1 

PLOS ONE

Dear Dr. Kitil, 

I'm pleased to inform you that your manuscript has been deemed suitable for publication in PLOS ONE. Congratulations! Your manuscript is now being handed over to our production team.

Kind regards, 

on behalf of

Dr. Tamirat Getachew 

Academic Editor

PLOS ONE